# A Novel Method for the Determination of Squalene, Cholesterol and Their Oxidation Products in Food of Animal Origin by GC-TOF/MS

**DOI:** 10.3390/ijms25052807

**Published:** 2024-02-28

**Authors:** Małgorzata Czerwonka, Agnieszka Białek, Barbara Bobrowska-Korczak

**Affiliations:** 1Department of Toxicology and Food Science, Faculty of Pharmacy, Medical University of Warsaw, Banacha 1, 02-097 Warsaw, Poland; barbara.bobrowska@wum.edu.pl; 2School of Health and Medical Sciences, University of Economics and Human Sciences in Warsaw, Okopowa 59, 01-043 Warsaw, Poland; a.bialek@vizja.pl; 3The Kielanowski Institute of Animal Physiology and Nutrition, Polish Academy of Sciences, Instytucka 3, 05-110 Jabłonna, Poland

**Keywords:** cholesterol, squalene, cholesterol oxidation products, food of animal origin, 7-ketocholesterol, gas chromatography, mass spectrometry

## Abstract

Cholesterol present in food of animal origin is a precursor of oxysterols (COPs), whose high intake through diet can be associated with health implications. Evaluation of the content of these contaminants in food is associated with many analytical problems. This work presents a GC-TOF/MS method for the simultaneous determination of squalene, cholesterol and seven COPs (7-ketocholesterol, 7α-hydroxycholesterol, 7β-hydroxycholesterol, 25-hydroxycholesterol, 5,6α-epoxycholesterol, 5,6β-epoxycholesterol, cholestanetriol). The sample preparation procedure includes such steps as saponification, extraction and silylation. The method is characterized by high sensitivity (limit of quantification, 0.02–0.25 ng mL^−1^ for instrument, 30–375 μg kg of sample), repeatability (RSD 2.3–6.2%) and a wide linearity range for each tested compound. The method has been tested on eight different animal-origin products. The COP to cholesterol content ratio in most products is about 1%, but the profile of cholesterol derivatives differs widely (α = 0.01). In all the samples, 7-ketocholesterol is the dominant oxysterol, accounting for 31–67% of the total COPs level. The levels of the other COPs range between 0% and 21%. In none of the examined products are cholestanetriol and 25-hydroxycholesterol present. The amount of squalene, which potentially may inhibit the formation of COPs in food, ranges from 2 to 57 mg kg^−1^.

## 1. Introduction

Cholesterol is the main sterol in food products of animal origin. For years, its high consumption was considered one of the causes of atherosclerosis. Now it is known that even a significant amount of this compound in the diet does not have health implications [1]. However, cholesterol is susceptible to the oxidation process, which makes it a precursor of oxysterols [2].

Cholesterol oxidation products (COPs) are formed in foods during processing and storage [3]. They are compounds which, in addition to the hydroxy group substituted on the third carbon, have an additional hydroxy, ketone or epoxide group on the steroid core and/or hydroxy group on the side chain (Figure 1) [4]. In the animal body, they are intermediates in the synthesis of bile acids, steroid hormones or 1,25-(OH)-cholecalciferol, and they perform many regulatory functions [5]. They show pro-inflammatory and proapoptotic effects [6]. However, too high levels of these compounds in the body are associated with increased oxidative stress, inflammatory processes, and risk of diseases such as cancer, cardiovascular or neurodegenerative disease [7,8]. The source of these compounds in the human body is endogenous synthesis and food of animal origin. COPs are better absorbed from the gastrointestinal tract than cholesterol itself and have a stronger effect on the organism than oxidized derivatives of plant sterols [9]. They are natural food contaminants, but the safe minimum limits of these compounds have not been established yet. The food contains cholesterol derivatives with an additional group substituted on the sterol ring, such as 7-ketocholesterol (7K), 7α-hydroxycholesterol (7αOH), 7β-hydroxycholesterol (7βOH), 5,6β-epoxycholesterol (5,6βE), 5,6α-epoxycholesterol (5,6αE) and cholestanetriol (Triol) [10]. Their content in products of animal origin usually does not exceed 1% of the total content of sterol compounds; however, in some products, COPs’ share may reach 10% [11]. Because the level of COPs in food increases during long storage and technological processes such as heat treatment or chopping [12], these compounds can be also used as quality markers [13].

Squalene is a polyunsaturated triterpene widely found in nature and produced by plant, animal, fungi and bacteria cells [14]. In living organisms, it is a cholesterol precursor. Squalene is a valuable component of the diet of comprehensive biological function [15]. Studies have shown its chemopreventive, antibacterial, and anti-fungal properties. Also, its beneficial effect has been demonstrated in the prevention of vascular diseases [16,17]. Food of animal origin is usually not a very good source of squalene, but due to antioxidant activity, this compound can prevent the oxidation of lipid components, including cholesterol, and prevent the formation of COPs [18].

Determination of cholesterol and squalene, due to the relatively high content in food, does not pose significant analytical problems [19,20]. However, the topic of the cholesterol oxidation derivatives analysis is relatively new, and the contents of these compounds are several dozen hundred times lower than the parent compound [21]. There is currently no reference procedure for COP analysis, and the most common method is to use gas or liquid chromatography with mass detection [22]. However, emerging reports on the contents and profiles of COPs in food are very divergent, which may result from methodological differences [23]. The sample preparation seems to be the most crucial step in COP analysis. Most often, this process involves the following steps: extraction of lipids from the food matrix, hydrolysis of esterified sterols, and purification on SPE columns or liquid–liquid extraction [24]. However, each of the above stages can be carried out in a very different way, which can significantly affect the result of the analysis. It seems necessary to create a uniform, relatively simple method for COP analysis in food. This work aimed to create a simultaneous procedure of squalene, cholesterol and cholesterol oxidation product determination and exploit it to evaluate the content of these substances in selected food products of animal origin.

## 2. Results

This work resulted in an efficient procedure for the simultaneous determination of squalene, cholesterol and seven cholesterol oxidation products by GC-TOF/MS. Sample preparation was based on saponification, extraction and derivatization with BSTFA (N,O-Bis(trimethylsilyl)trifluoroacetamide). The exact procedure is described in the Section 4. Figure 2 shows chromatographic separation, while Table 1 presents the retention times of the determined compounds and the main diagnostic ions. The validation parameters of the method are shown in Table 2. The method was characterized by low limits of detection (LOD) and quantification (LOQ) of the determined compounds. The LODs ranged from 0.01 (7αOH, 7βOH, Triol) to 0.08 (squalene) ng μL^−1^ for the instrument and from 30 to 375 μg kg^−1^ of the real sample. The determined linearity range of the method was also high, especially for cholesterol (max 500 ng μL^−1^). Recovery was above 85% for all the compounds. The repeatability, expressed as a relative standard deviation (RSD), was no higher than 6.2%.

The method was verified by determining the squalene, cholesterol and COP content of eight animal products of varied origin, diversified matrices and diverse total fat content (milk, edam cheese, yoghurt, eggs (yolk), chicken breast meat, pork chop, salmon, cod). In the case of egg yolk, there was a need to dilute the sample injected into the GC-TOF/MS and determine cholesterol separately because the peak was too saturated. The results are shown in Table 3. The content of the tested compounds varied greatly between the products (ANOVA, α = 0.01). The squalene content ranged from 2.7 (milk) to 57 mg kg^−1^ (egg yolk). The cholesterol levels varied between 60 and over 14,500 mg kg^−1^, while the sum of its oxidized derivatives was from 0.7 to 32 mg kg^−1^. In the present work, five oxidized cholesterol derivatives were determined. In most examined food samples, 7K, 7αOH, 7βOH, 5,6αE, and 5,6βE were present. These product groups were characterized by significantly different amounts and profiles of the COPs (ANOVA, α = 0.01). The predominant oxysterol was 7-ketocholesterol, whose content ranged between 31 (chicken breast meat) and 67% (egg yolk) of all the COPs. The content of the remaining COPs ranged from 0 to 29% (Figure 3). Moreover, 7αOH was not found in the egg yolk and 5,6αE was below the limit of detection in the egg yolk and yoghurt. In none of the samples, 25OH and Triol were present. The ratio of cholesterol oxidation products to their precursor concentration is, on average, about 1%.

## 3. Discussion

### 3.1. Method for Squalene, Cholesterol and Oxidized Cholesterol Derivative Determination in the Food of Animal Origin Using GC-TOF-MS

Cholesterol determination is commonly performed in food-testing laboratories. Multiple analytical methods have been developed for the analysis of this compound, including classical chemical methods, enzymatic assays, gas chromatography (GC) and liquid chromatography (LC) connected with various detectors, including mass spectrometers [25,26]. However, the determination of oxidized cholesterol derivatives, mainly due to their low content in food, is more demanding [23]. So far, there is no reference analytical method for this group of compounds. Gas chromatography with mass spectrometry is the most popular method, though the sample preparation procedure is the most problematic stage [21]. Determination of the squalene content in food products is not a large research challenge. Most often, liquid chromatography with UV detection or mass spectrometry is used for this purpose [19]. The procedure using gas chromatography is carried out on the occasion of analysis of the fatty acids methyl esters or cholesterol and other sterol compounds [27,28]. Analytical methods aimed at only squalene content determination are rare. In our study, the information on the squalene content is very valuable because this compound, as an antioxidant, can potentially inhibit the formation of COPs.

The first step in the sample preparation procedure was to add an internal standard and BHT (butylated hydroxytoluene), as an antioxidant, to the weighted sample. Here, 5α-cholestane is the most commonly used internal standard in sterol compounds analysis in animal material [29]. Then, KOH in ethanol was added. Most samples did not require fat extraction and direct saponification was performed. The exceptions were the milk and yoghurt samples. The KOH concentration was quite high (1 M); however, it did not cause losses of the tested compounds. Moreover, it affected the structure of the food, facilitated homogenization and thus extraction of analytes from the food. Saponification was carried out at room temperature for 20–22 h. During the preliminary study, different hydrolysis times and temperatures and the addition of different antioxidants at different levels were investigated. The presented procedure is the optimum obtained from our experiments. In other publications, saponification at elevated temperature, lasting several dozen minutes, is quite common practice but can cause a change in the quantity and profile of these COPs and thus falsify the results [22].

Cholesterol and oxysterols are present in food in two forms: free and bound. The majority of sterols are present in free form, and only 10–15% of them are esterified, most commonly with fatty acids [20,30]. During saponification, sterol compounds are released from ester linkages [31] and the total content of cholesterol and its derivatives is determined. Several studies excluded the hydrolysis/saponification stage, while the sample was purified using SPE [22]. This method is incomparably faster, but it enables the determination of free forms of sterols only. In our digestive tract, sterols are released from ester bonds by cholesterol esterase [30], which is why it seems appropriate to determine the total content of sterols in food.

The next stage of the sample preparation was liquid–liquid extraction with hexane. This process was characterized by very high efficiency. The concentration of the sample by evaporation of hexane under a stream of nitrogen could have been the last stage of the sample preparation. It is possible to determine cholesterol and its derivatives using gas chromatography without the derivatization process [32]. However, it was not possible to separate the 7-hydroxycholesterol anomers with the used column without silylation. In addition, the detection and quantification limits were improved after derivatization. 

The silylation process was carried out using BSTFA in pyridine at a temperature of 80 °C and a time of 40 min. From an economic and ecological point of view, it was important to reduce the amounts of reagents during this process. After the derivatization process, the sample was diluted with hexane and applied to the column. 

The presented method is not free of defects. The biggest problem is the huge difference between the cholesterol and oxysterol concentrations in the food samples, so only the internal standard and cholesterol are visible on the main chromatogram (total ion chromatogram, TIC), whereas the individual COPs are revealed only by the monitoring of the respective ions (Appendix A). The peak of cholesterol almost never resembled the shape of a Gaussian curve; it was always a little overloaded. In the case of egg yolk (weighing about 70–80 mg), it was not possible to simultaneously determine cholesterol and its oxidized derivatives. It was necessary to dilute the sample and re-inject it into the column. By using larger weights of samples, and larger amounts of reagents, it is possible to improve the limits of detection and determination for COPs in food products. However, this would require additional chromatographic separation solely for the cholesterol content.

### 3.2. Evaluation of Squalene, Cholesterol, and Cholesterol Oxidation Product Content in Selected Food Products of Animal Origin

The study of squalene, cholesterol and COPs in a selection of eight food groups was designed to validate the new method, test it in a variety of matrices and identify further directions for our research.

The squalene content determined in this study is similar to the values presented in other publications [33,34]. Squalene is a compound commonly present in foods of both plant and animal origin. Its levels in amaranth or olive oils reach a few g per kilogram of products [19]. Therefore, meat, sausages, fish, eggs and dairy products are not the best sources of this compound in the diet. However, its content in animal food products seems to be important due to the widely described strong antioxidant properties [35]. 

Cholesterol is a sterol formed from squalene only in animal organisms [36]. The content of this compound in foods ranges widely from a few dozen mg per kg in lean dairy products (milk, yoghurt) to a dozen g per kg in egg yolk. It seems that high cholesterol intake through a diet has no health implications [1]. This compound, however, undergoes the oxidation process and is a precursor of oxysterols, which, when taken in large amounts along with a diet, may contribute to an increased risk of many non-communicable diseases [1,6].

In this study, 7K was the main COP in all the food products. High levels of this compound relative to the total COP concentration were also presented by other authors [3,13,36]. Some studies suggest that 7K can be used as an indicator of cholesterol oxidation [37,38]. However, its share of the overall content of COPs in food varies within a very wide range. The food matrix may affect COP formation [39]. Therefore, the analysis of the content and profile of all the cholesterol derivatives, not only 7K, seems to be the more appropriate approach.

Two COPs were not detected in the tested food samples: Triol and 25OH. Triol is a compound formed from epoxides and is a further product of the cholesterol transformation [40]. Moreover, 25OH is a cholesterol derivative that has an additional hydroxyl group attached to the side chain, not the core. Typically, oxysterols of this type are formed by enzymatic rather than free radical reactions [41], which may explain the absence (levels below detection) of 25OH in foods.

The ratio of cholesterol oxidation products to their precursor concentration is on average about 1%. According to other authors, this value is most common, but in some foods, it can be higher [11]. Meanwhile, in egg yolk, despite the highest cholesterol and oxysterols content, the COP to cholesterol ratio was relatively low (0.2%). The results obtained in this study do not indicate a simple relationship between the oxysterol amount and the level of lipid compounds or cholesterol itself. The process of non-enzymatic oxidation of cholesterol in food is affected by several factors, both promoting and inhibiting its course [42]. In the case of meat, fish and their products, some of the COPs probably were formed naturally in the animal’s body. The time and conditions of food processing and/or storage also affect the oxidation of fat components, including cholesterol [4]. The composition of the product is also important [36]. The total fat content and fatty acid profile may affect cholesterol oxidation [43]. Squalene, as an antioxidant, also can prevent COP formation. However, based on this study, it is not possible to estimate the impact of the squalene content on the cholesterol oxidation process. Further work in this area is necessary, carried out in the first place on model systems.

The levels of COPs examined in this study are similar to the results presented by other authors [10,20,44]. However, it is quite difficult to compare the content of oxidized cholesterol derivatives determined in this work with the data presented by other authors. This is the result of different determination methods, especially sample preparation procedures. The lack of a reference method means that the available information is diversified, sometimes contradictory, and it is not possible to draw general conclusions based on the available data. One of the possible solutions may be simultaneous determination of the COP content in the same product using different analytical procedures and different methods of determination. This will allow also to assess the reproducibility of each of the methods. Another possibility may be multicenter comparative studies performed in the same analytical samples (proficiency testing), which allow access to the compatibility of different methods. The next approach might be the re-validation of the applied analytical methods with subtle changes in the examined conditions based on the available analytical data, which will indicate the most pivotal parameters.

## 4. Materials and Methods

### 4.1. Reagents and Research Material

Analytical standards of cholesterol and COPs, 5α-cholestane (internal standard, HPLC grade), silylation mixture (BSTFA + 1% TMCS (N,O-Bis(trimethylsilyl)trifluoroacetamide with 1% of trimethylchlorosilane, for GC derivatization) and pyridine (HPLC grade) were bought from Sigma Aldrich Corp., Poznań, Poland. Ethanol (96%, analytical grade), methanol (HPLC grade), hexane (HPLC grade), chloroform (analytical grade), potassium hydroxide (analytical grade), and butylated hydroxytoluene (BHT, analytical grade) were purchased in Avantor Performance Materials Poland S.A., Gliwice, Poland. Nitrogen (purity: ≥99.999%) and helium (purity: ≥99.9999%) were provided by Air Products and Chemicals, Inc., Warsaw, Poland.

The research material was selected animal products purchased in supermarkets in Warsaw: milk, Edam cheese, yoghurt, eggs, chicken breast meat, pork chop, salmon, and cod. All the products were stored at fridge temperature (2–3 °C) and were tested before the expiry date.

### 4.2. Determination of Squalene, Cholesterol and Cholesterol Oxidation Products in Food Samples by Gas Chromatography–Time of Flight–Mass Spectrometry Method

#### 4.2.1. Sample Preparation

The sample preparation procedure included such steps as (a) saponification/hydrolysis, (b) liquid–liquid extraction and (c) derivatization. Each food sample (50–300 mg depending on fat content) or extracted fat (40–50 mg), after the addition of 25 µL of internal standard solution (5α-cholestane, 0.5 mg mL^−1^ in hexane) and 10 µL of BHT solution (5 mg mL^−1^ in ethanol), was homogenized in 3 mL 1 M KOH in ethanol. The hydrolysis lasted 20–22 h at room temperature (20–22 °C). Then, 4 mL of water and 2 mL of hexane were added to the sample and shaken vigorously. The hexane layer was collected into a 2 mL vial and then evaporated under a stream of nitrogen. Next, 50 µL of pyridine and 25 µL of the silylation mixture were added to the dry residue and mixed very thoroughly. Derivatization was carried out at 80 °C for 40 min. Then, 225 µL of hexane was added to the sample, mixed and injected into the GC column.

#### 4.2.2. Chromatographic Separation and Detection

Chromatographic separation and detection were performed using GC-TOF/MS (Pegasus^®^ BT, LECO Corporation, St. Joseph, MI, USA) on an Rxi^®^-17SilMS (30 m × 0.25 mm × 0.25 μm, Restek, Bellefonte, PA, USA) column. First, 1 µL of the sample (splitless mode) was applied to the column. The injector temperature was 290 °C. The temperature program was as follows: 200 °C–4.60 min; increase 5 °C per min to 290 °C; 290 °C for 12.4 min. The carrier gas (helium) flow was 1 mL min^−1^. The transfer line temperature was set to 290 °C. EI ionization was used (temperature: 250 °C, energy: 70 eV). Compounds were identified based on the retention time and mass spectra (presented in additional materials), quantitative analysis based on standard curves determined for each compound and internal standard.

#### 4.2.3. Validation Parameter

The procedure described above is the optimized version. During the work, such parameters as the KOH and BHT concentration, time and conditions of saponification, the necessity of derivatization, time, and conditions of silylation were checked. The procedure was fully validated according to procedures described by Simoneau et al. [45]. The linear regression coefficient (R^2^) was determined for each calibration curve. Repeatability was presented as the relative standard deviation (RSD) for one sample (*n* = 8). Laboratory-made spiked samples were used to determine the recovery. Three levels of standard solutions additive were used: small, medium and large, which for each test compound corresponded to 25, 50 and 75% of the maximal concentration determined in the range of linearity (*n* = 6 per each level). The repeatability and recovery were determined using an Edam cheese sample. The limit of detection (LOD) and limit of quantification (LOQ) were calculated based on the signal-to-noise ratio, which was respectively ≥3 and ≥10.

### 4.3. Fat Content Determination

The fat content in the food of animal origin samples was determined gravimetrically after extraction with a mixture of chloroform/methanol (*v*/*v* 2:1) and solvent evaporation under a stream of nitrogen. The procedure was described by Folch et al. [46]

### 4.4. Statistical Analysis

All the analyses of animal food products were performed in duplicate. The results were evaluated using STATISTICA 13.3 software (StatSoft, Krakow, Poland) and presented as the mean values ($\bar{x}$) and standard deviations (SD). The repeatability of the method was given as the relative standard deviation (RSD). The normality of the data within groups was determined using the Shapiro–Wilk test. One-way ANOVA (α = 0.01) followed by Tukey’s test (α = 0.05) was used to analyze the normal distribution data. The Kruskal–Wallis test (α = 0.05) followed by a post hoc test for multiple comparisons of the mean rank (α = 0.05) was used for the non-parametric data. The same letters indicate homogeneous groups.

## 5. Conclusions

Oxidized cholesterol derivatives are natural food contaminants. Their high intake along with the diet can contribute to increased levels of these compounds in the human body and thus contribute to many health problems. Although the topic of oxidized cholesterol derivatives in food is not new, there is no reference analytical method for their determination. Differences in sample preparation procedures cause the results obtained by different research teams to be very diverse. This paper proposes a method for COPs’ analysis and presents their content in eight groups of food products. The large difference between the cholesterol and cholesterol derivatives content in the foods causes the COPs to be identified only by the monitoring of the respective ions, usually not visible at total ion chromatogram, which is a limitation of this method.

Evaluation of the exact content of oxysterols in food will allow us to estimate the intake of these compounds along with the diet and develop the limits in foodstuffs. Thorough knowledge of the factors determining their content, like technological or storage parameters, or related to the composition of the product itself (including squalene content), will enable the development of mitigation strategies for these contaminants in food. However, to achieve this, reliable analytical tools are necessary.

## Figures and Tables

**Figure 1 ijms-25-02807-f001:**
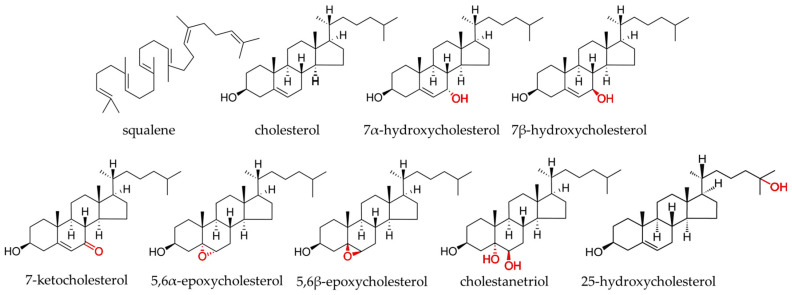
Structural formulae of squalene, cholesterol, and their oxidized derivatives.

**Figure 2 ijms-25-02807-f002:**
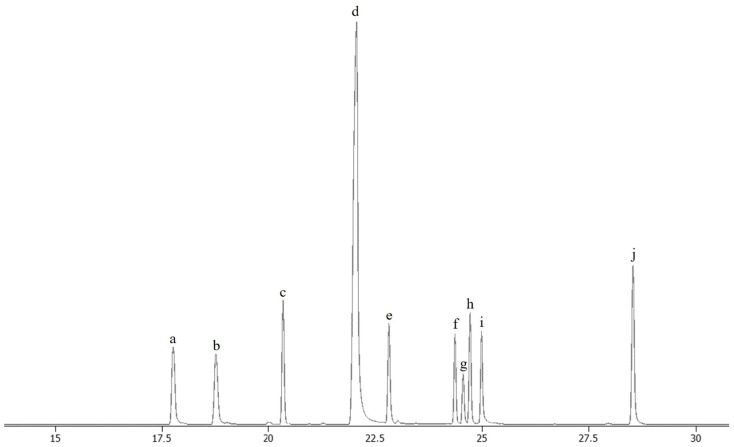
Chromatographic separation of analytical standards of squalene, TMS derivatives of cholesterol and cholesterol oxidation products by GC-TOF/MS (total ion chromatogram). a—squalene; b—internal standard (5α-cholestane); c—7α-hydroxycholesterol TMS; d—cholesterol TMS; e—7β-hydroxycholesterol TMS; f—5,6β-epoxycholesterol TMS; g—cholestanetriol TMS; h—5,6α-epoxycholesterol TMS; i—25-hydroxycholesterol TMS; and j—7-ketocholesterol TMS. TMS—trimethyl silane derivative.

**Figure 3 ijms-25-02807-f003:**
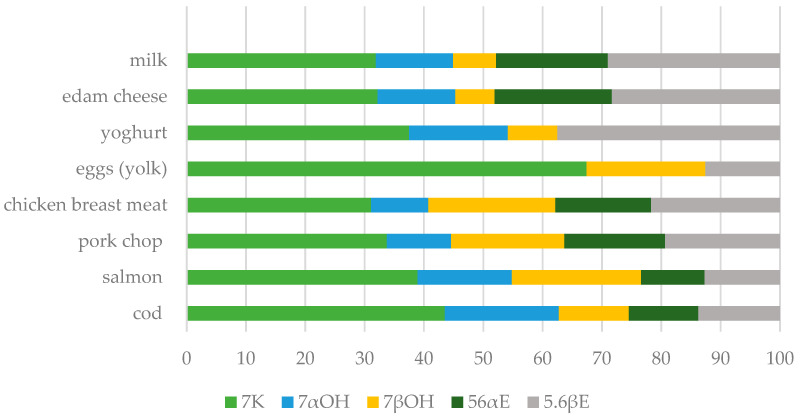
Profile of the cholesterol oxidation products in selected products of animal origin (mean values): 7K—7-ketocholesterol; 7αOH—7α-hydroxycholesterol; 7βOH—7β-hydroxycholesterol; 5,6αE—5,6α-epoxycholesterol; and 5,6βE—5,6β-epoxycholesterol.

**Table 1 ijms-25-02807-t001:** Identification parameters for squalene, cholesterol, and cholesterol oxidation products determined by the GC-TOF/MS method.

Compound	Rt	Molecular Weight	Main Diagnostic Ions
	[min]	[Da]	[Da]
Sq	17.81	410.7	410.4
CH-TMS	22.24	458.8	386.4, 458.4
7αOH-TMS	20.41	549.0	546.5, 456.5
7βOH-TMS	22.82	549.0	546.5, 456.5
5,6βE-TMS	24.22	474.8	384.3, 474.4
Triol-TMS	24.42	546,0	456.5, 546.5
5,6αE-TMS	24.66	474.8	384.3, 474.4
25OH-TMS	25.08	547.0	547.0
7K-TMS	27.98	472.8	472.3

Rt—retention time; Sq—squalene; CH—cholesterol; 7K—7-ketocholesterol; 7αOH—7α-hydroxycholesterol; 7βOH—7β-hydroxycholesterol; 5,6αE—5,6α-epoxycholesterol; 5,6βE—5,6β-epoxycholesterol; 25OH—25-hydroxycholesterol; Triol—cholestanetriol; TMS—trimethylsilyl derivatives.

**Table 2 ijms-25-02807-t002:** Validation parameters of the GC-TOF/MS method for squalene, cholesterol and cholesterol oxidation products.

Compound	LOD Instrument *	LOQ Instrument *	LOD Method **	LOQ Method **	The Range of Linearity	Repeatability (*n* = 8)	Recovery *** (*n* = 8)
	[ng μL^−1^]	[ng μL^−1^]	[μg kg^−1^]	[μg kg^−1^]	[ng μL^−1^]	RSD [%]	$\bar{x}$ ± SD [%]
Sq	0.08	0.25	120	375	0.25–70	3.4	90.6 ± 1.16
CH	0.06	0.20	90	300	0.20–500	2.7	94.5 ± 1.21
7αOH	0.01	0.03	15	45	0.03–10	4.2	89.3 ± 1.26
7βOH	0.01	0.03	15	45	0.03–10	3.2	89.7 ± 0.62
5,6βE	0.05	0.16	75	240	0.16–10	2.8	92.1 ± 1.05
Triol	0.01	0.02	15	30	0.02–2	6.2	87.4 ± 2.03
5,6αE	0.02	0.06	30	90	0.06–10	4.3	89.1 ± 2.31
25OH	0.04	0.12	60	180	0.12–10	5.2	85.3 ± 1.42
7K	0.02	0.06	30	90	0.06–10	2.3	95.3 ± 1.01

* Limit of detection (LOD, ≥3× S/N) and limit of quantification (LOQ, ≥10× S/N) for the instrument (GC-TOF/MS). ** Limit of detection (LOD) and limit of quantification (LOQ) for the method, sample size: 200 mg (Edam cheese). *** Recovery—based on fortified samples of Edam cheese. Sq—squalene; CH—cholesterol; 7K—7-ketocholesterol; 7αOH—7α-hydroxycholesterol; 7βOH—7β-hydroxycholesterol; 5,6αE—5,6α-epoxycholesterol; 5,6βE—5,6β-epoxycholesterol; 25OH—25-hydroxycholesterol; Triol—cholestanetriol.

**Table 3 ijms-25-02807-t003:** Squalene (Sq), cholesterol (CH) and cholesterol oxidation products (COPs) content in selected food products of animal origin expressed as the mean and (standard deviation).

Products	n	Fat	Sq	CH	7K	7αOH	7βOH	5,6αE	5,6βE	∑ COPs	COPs/CH
$\bar{x}$ (SD)		[%]			[mg kg^−1^]						[%]
Milk *	4	3.35 (0.30)	2.68 (0.39)	63.99 (7.22)	0.22 (0.03)	0.09 ^a^ (0.02)	0.05 ^a^ (0.01)	0.13 ^a^ (0.03)	0.20 (0.03)	0.65 (0.1)	1.01 (0.06)
Edam cheese	6	26.41 (1.51)	19.39 (4.34)	451.29 (50.26)	1.84 (0.19)	0.75 (0.04)	0.38 (0.04)	1.13 (0.17)	1.62 (0.11)	5.60 (0.3)	1.25 (0.16)
Yoghurt *	6	2.98 (0.10)	2.99 (1.16)	60.51 (9.05)	0.27 ^a^ (0.12)	0.12 ^b^ (0.06)	0.06 ^b^ (0.02)	nd	0.27 ^a^ (0.11)	0.72 (0.26)	1.20 (0.40)
Eggs (yolk)	8	29.27 (4.85)	56.98 (7.88)	14,562.5 (537.0)	23.51 ^a^ (20.72)	nd	6.96 ^a,b^ (7.75)	nd	4.39 ^b^ (0.64)	32.24 (19.49)	0.22 (0.13)
Chicken breast meat	8	2.13 (0.17)	8.41 (1.45)	483.4 (122.2)	1.29 (0.23)	0.40 (0.08)	0.89 ^a^ (0.10)	0.67 (0.1)	0.90 ^a^ (0.11)	4.15 (0.32)	0.94 (0.40)
Pork chop	8	3.24 (0.71)	10.85 (3.41)	525.0 (71.0)	1.43 (0.33)	0.46 (0.09)	0.81 ^a^ (0.13)	0.72 ^a^ (0.13)	0.82 ^a^ (0.10)	4.24 (0.62)	0.81 (0.05)
Salmon	6	8.96 (2.70)	32.64 (8.72)	497.3 (51.3)	1.93 (0.07)	0.79 ^a^ (0.05)	1.08 ^a^ (0.38)	0.53 ^a^ (0.03)	0.63 ^a^ (0.01)	4.97 (0.41)	1.01 (0.12)
Cod	6	1.31 (0.25)	10.45 (3.54)	513.8 (99.54)	3.13 (0.54)	1.38 ^a^ (0.29)	0.85 ^b^ (0.09)	0.84 ^b^ (0.15)	0.99 ^a,b^ (0.16)	7.19 (1.19)	1.47 (0.49)

7K—7-ketocholesterol; 7αOH—7α-hydroxycholesterol; 7βOH—7β-hydroxycholesterol; 5,6αE—5,6α-epoxycholesterol; 5,6βE—5,6β-epoxycholesterol; Triol—cholestanetriol. n—number of samples of different brands. nd—not detected. ^a,b^—homogeneous group in rows; comparison of COPs content in products (Tukey’s test, α = 0.05/* multiple comparisons of mean rank (non-parametric data), α = 0.05).

## Data Availability

The data that support the findings of this study are available from the corresponding author upon reasonable request.

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
