# Peer review of "A Novel Method for the Determination of Squalene, Cholesterol and Their Oxidation Products in Food of Animal Origin by GC-TOF/MS"

_ijms, 2024, doi:10.3390/ijms25052807_

Round 1

Reviewer 1 Report

Comments and Suggestions for Authors

  • A brief summary- The manuscript reports a GC-TOF/MS method that may be used for the measurement of oxysterols, squalene, and cholesterol in foods of animal origin. The study provides a detailed description of the method along with information about any issues encountered during its development. As such it may provide a solid base for the development of a standard methodology in the future or a direct use of the presented method in any laboratory equipped with a similar instrument.
  • General concept comments- The presented study may find some interest among the scientists that require such a method for their research. Furthermore, once threshold toxicity values for oxysterols are established, such methodologies may be in high demand among analytical laboratories.
  • In the opinion of the reviewer, the article requires minor revision. The paper is in general well written, and there are only some minor issues that require additional explanation, change of words or perhaps change of the approach (as e.g. expression of the results for statistical data evaluation). These issues were mentioned in the specific comments.
  • Specific comments 
    • Title- reflects the content of the study very well but it would be good to also emphasize the element of novelty in it e.g. a novel method
    • Abstract- Describes the content of the study well. However, since the focus is on a novel method the abstract could give some more information about its performance e.g. limit of detection, quantification, uncertainties of the measurement; whatever was assessed. Lines 22-23- please add that the % refers to COPs content or whatever else the % value could mean.
    • Introduction- This section introduces the rationale behind the study well. To slightly improve it, the reviewer suggests linking the part about COPs and squalene better and addressing some comments below.

Lines 40-41, 61, 71-72- please provide references to the sentence    

Lines 63-64- if so, then please justify why to measure squalene in animal foods

Lines 78-81- some of the already mentioned information is given again, consider moving this fragment where information about COPs is given and removing repetition

    • Results- This section presents results concisely and clearly. There are only some very minor issues with the text mentioned below.

Lines 89- please, explain the abbreviation

Line 97- please mention table 2 before describing results contained in it

    • Discussion- This section is well-structured and contains an appropriate number of references to provide a good discussion. However, a comparison of the validation parameters achieved for the described method with other methods that exist in the literature could be added, as this would help to emphasize the impact of the publication. In addition, some specific comments that suggest only very minor corrections to improve the content of the section are given below.

Line 150- it would be better to mention these standard detectors as MS is quite a standard methodology nowadays as well.

Line 155- hence, it would be nice if the study provided explanation on why the authors decided to include squalene

Line 161- please, explain the abbreviation

Lines 180-181- this sentence gives a strange impression that authors are not sure If the extraction was performed properly. Please rephrase or add explanation.

Lines 182-183- could or was?

Line 183- starts sentence about the derivatization, should it not start a new paragraph?

Line 185-186- under what conditions was it not possible to separate the 7-hydroxycholesterol anomers? Without derivatisation? Was such an attempt made? Please, explain.

Line 187- please, explain abbreviations

Line 204- phrase “test the matrix effect assay” not clear, please rephrase

Lines 209-210- it would be of interest to read whether the levels of squalene found in foods in the reviewed manuscript were similar to these reported in foods of animal origin in other publications

Lines 212-214, 218-219- please provide a reference to the sentence

Lines 231-232- please, mention some exemplary foods where other authors found this proportion exceptionally high

Lines 240-241- what kind of fatty acid profile prevents cholesterol from oxidising? Please, explain.

Lines 254-255- laboratories call this proficiency testing. Alternatively, the term “round robin” could be used.

    • Materials and methods- since this section does not appear after the introduction, all the abbreviations that are explained in it should be explained earlier, after the first use in the text.

Line 295- there was no supplementary material available with the manuscript for the review, please ensure that it is visible, once the article is published

Line 317- please change the word recurrence to one reflecting the validation parameter that was assessed. It would be best to use a term defined by ISO.   

Section 4.4- since there is no mention of what exact data were analysed using which statistical test, the authors should mention selected tests in the footnotes of relevant tables and when describing results. In addition, for the comparison of the profiles of COPs as given in Figure 3(mentioned in the results but not clear if this was the format of data introduced to statistical evaluation), please bear in mind that pairwise statistics should be used after compositional data are transformed as e.g. recommended by Aitchison (Aitchinson, John. ‘The Statistical Analysis of Compositional Data’. The Statistical Analysis of Compositional Data J. Aitchison Journal of the Royal Statistical Society. Series B (Methodological) 44, no. 2 (1982): 139–77.).  

    • Conclusions- are appropriate and include a well written part about perspectives for the application of the developed method and further research.
    • Figure and Tables- All means of result expression are clear and appropriate in terms of presentation and description. The only change that may be required is in Table 2- is recurrence in fact repeatability?
    • References- The list contains an appropriate number of references (45) to justify the study and provide a discussion. The major part of the list includes references older than 5 years, however, this is justified by the subject that the study refers to. Authors should double-check the list as some of the references seem to be inconsistent in terms of style (no. 12, 19, and 34 are some that the reviewer has noticed).

Comments on the Quality of English Language

The quality of English is good and minor issues that are present do not affect the understanding of the text.

Author Response

Dear Reviewer,

Thank you very much for reviewing our article and your very valuable comments, which helped us to improve our manuscript. Your comments and our responses to them are listed below. We are very grateful for your time.

R: Title- reflects the content of the study very well but it would be good to also emphasize the element of novelty in it e.g. a novel method

A: Thank You very much for this suggestion. Our new title is ‘Novel method for the determination of squalene, cholesterol and their oxidation products in food of animal origin by GC-TOF/MS method’.

R: Abstract- Describes the content of the study well. However, since the focus is on a novel method the abstract could give some more information about its performance e.g. limit of detection, quantification, uncertainties of the measurement; whatever was assessed.

A: LOQs for the instrument and for the samples as well as the repeatability range have been added.

R: Lines 22-23- please add that the % refers to COPs content or whatever else the % value could mean.

A: Corrected as suggested by the Reviewer.

R: Lines 40-41, 61, 71-72- please provide references to the sentence

A:  References have been added.

R: Lines 63-64- if so, then please justify why to measure squalene in animal foods

A: We added: ‘…and prevent the formation of COPs’

R: Lines 78-81- some of the already mentioned information is given again, consider moving this fragment where information about COPs is given and removing repetition

A: We agree with the Reviewer. The part where the information was repeated has been deleted.

R: Lines 89- please, explain the abbreviation

A: The abbreviation has been explained.

R: Line 97- please mention table 2 before describing results contained in it

A: This sentence has been moved above information presented in Table 2.

R: Line 150- it would be better to mention these standard detectors as MS is quite a standard methodology nowadays as well.

A: We changed: ‘various detectors, including mass spectrometers’

R: Line 155- hence, it would be nice if the study provided explanation on why the authors decided to include squalene

A: We added: ‘In our study, the information on squalene content is very valuable because this compound, as an antioxidant, can potentially inhibit the formation of COPs.’

R: Line 161- please, explain the abbreviation

A: Abbreviation has been explained

R: Lines 180-181- this sentence gives a strange impression that authors are not sure If the extraction was performed properly. Please rephrase or add explanation.

A: This step caused the most problems for less experienced analysts, especially the collection of the hexane phase. However, we agree with the reviewer that this information can be confusing. We changed: ‘This process was characterized by very high efficiency.’

R: Lines 182-183- could or was?

A: ‘Could’. This could have been the final step (perhaps with a different column), but it is not. We explained this in the following sentences.

R: Line 183- starts sentence about the derivatization, should it not start a new paragraph?

A: We agree. It has been changed to a separate paragraph.

R: Line 185-186- under what conditions was it not possible to separate the 7-hydroxycholesterol anomers? Without derivatisation? Was such an attempt made? Please, explain.

A: We changed this sentences: ‘However, it was not possible to separate the 7-hydroxycholesterol anomers with the used column without silylation. Besides, detection and quantification limits were improves after derivatization.’

R: Line 187- please, explain abbreviations

A: The BSTFA abbreviation has already been explained (line 90).

R: Line 204- phrase “test the matrix effect assay” not clear, please rephrase

A: We changed: ‘The study of squalene, cholesterol and COPs in a selection of eight food groups was designed to validate the new method, test it in a variety of matrices and chart further directions for our research.’

R: Lines 209-210- it would be of interest to read whether the levels of squalene found in foods in the reviewed manuscript were similar to these reported in foods of animal origin in other publications

A: When we wrote this manuscript, we wanted to compare squalene levels with data obtained by other authors. Unfortunately, there is very little work on the levels of this ingredient in foods of animal origin as usually, studies on squalene deal with products of plant origin. As a matter of fact we were able to compare our results with 2 or 3 other articles that indicated our results to be correct. Therefore, we have dropped this part. We totally agree with the Reviewer that it would have enriched the paper, but as we had too little data, we have decided to omit this part.

R: Lines 212-214, 218-219- please provide a reference to the sentence

A: The references have been added.

R: Lines 231-232- please, mention some exemplary foods where other authors found this proportion exceptionally high

A: We changed this sentence: ‘According to other authors, this value is most common, but in some foods, it can be higher [11].’

R: Lines 240-241- what kind of fatty acid profile prevents cholesterol from oxidising? Please, explain.

A: We changed this sentence to: ‘Total fat content and fatty acid profile may affect cholesterol oxidation [43].’

R: Lines 254-255- laboratories call this proficiency testing. Alternatively, the term “round robin” could be used.

A: We have added this term.

R: Materials and methods- since this section does not appear after the introduction, all the abbreviations that are explained in it should be explained earlier, after the first use in the text.

A: The problem with the abbreviations is that we wrote the article with the methodology before the results, and then adapted the manuscript to the journal's requirements (where the method is at the end). We apologise for this oversight. All necessary modifications have been implemented.

R: Line 295- there was no supplementary material available with the manuscript for the review, please ensure that it is visible, once the article is published

A: We added supplementary material with exemplary chromatogram and mass spectra of each compound.

R: Line 317- please change the word recurrence to one reflecting the validation parameter that was assessed. It would be best to use a term defined by ISO. 

A: We would like to thank for this remark. Term ‘recurrence’ has been replaced by ‘repeatability’ according to ‘ICH guideline Q2(R2) on validation of analytical procedures’.

R: Section 4.4- since there is no mention of what exact data were analysed using which statistical test, the authors should mention selected tests in the footnotes of relevant tables and when describing results. In addition, for the comparison of the profiles of COPs as given in Figure 3(mentioned in the results but not clear if this was the format of data introduced to statistical evaluation), please bear in mind that pairwise statistics should be used after compositional data are transformed as e.g. recommended by Aitchison (Aitchinson, John. ‘The Statistical Analysis of Compositional Data’. The Statistical Analysis of Compositional Data J. Aitchison Journal of the Royal Statistical Society. Series B (Methodological) 44, no. 2 (1982): 139–77.).  

A: We have checked the entire manuscript for the description of statistical analysis. All tests were used according to their assumptions and they were performed with the utmost care. We added proper descriptions to the individual figures and tables.

R: The only change that may be required is in Table 2- is recurrence in fact repeatability?

A: We would like to explain, that term ‘recurrence’ has been replaced by ‘repeatability’. moreover, the name of the statistical test has been added to Table 2 deciyption.

R: Authors should double-check the list as some of the references seem to be inconsistent in terms of style (no. 12, 19, and 34 are some that the reviewer has noticed).

A: Thank you very much for noticing the errors. References list has been checked and corrected.

Reviewer 2 Report

Comments and Suggestions for Authors

A well prepared and thought out research report. The area is of significant interest to researchers and the industry. The paper evaluates a range of products which are available for purchase and they use standard methods. Perhaps the authors could discuss how variations in levels do occur due to both differences in processing and ingredient selection. In particular, some of these products would have differences in composition through seasonality. How would that work ? 

Could you expand on the effects of plant and animal ingredients especially in plant based foods as opposed to animal foods, with reference to Vegan or Vegetarian diets compared to carbohydrate or meat avoidance diets ?  Would this be related to overall oil absorption levels so that we can look at food processing techniques in more detail ? 

Perhaps the autjhors could also have a discussion of processing of certain foods and extend the discussions already in the manuscript. I would suggest that the following references could be included as new but very applied focused research which has a potential effect on the food industry and the consumer, as well as how different oils are used in the food industry. 

Cui, N., Zhao, T., Han, Z., Yang, Z., Wang, G., Ma, Q. and Liang, L. (2022), Characterisation of oil oxidation, fatty acid, carotenoid, squalene and tocopherol components of hazelnut oils obtained from three varieties undergoing oxidation. Int J Food Sci Technol, 57: 3456-3466. https://doi.org/10.1111/ijfs.15669

Huang, Z., Du, M., Qian, X., Cui, H., Tong, P., Jin, H., Feng, Y., Zhang, J., Wu, Y., Zhou, S., Xu, L., Xie, L., Jin, J., Jin, Q., Jiang, Y. and Wang, X. (2022), Oxidative stability, shelf-life and stir-frying application of Torreya grandis seed oil. Int J Food Sci Technol, 57: 1836-1845. https://doi.org/10.1111/ijfs.15561

Shao, J., Huang, X., Liu, J. and Di, D. (2022), Characteristics and trends in global olive oil research: A bibliometric analysis. Int J Food Sci Technol, 57: 3311-3325. https://doi.org/10.1111/ijfs.15659

Martysiak-Å»urowska, D. and OrzoÅ‚ek, M. (2023), The correlation between nutritional and health potential and antioxidant properties of raw edible oils from cultivated and wild plants. Int J Food Sci Technol, 58: 676-685. https://doi.org/10.1111/ijfs.16217

Would it be possible to include some examples of food regulations in different parts of the world and how these issues are important in terms of overall food safety concerns and toxicity ? 

A very minor aspect I would ask the authors to consider is to emphasise the novelty of their research and how the research enhance current knowledge or practices. 

Author Response

Dear Reviewer,

Thank You for taking the time to review our paper. This manuscript deals only with the method we have developed for the simultaneous determination of squalene, cholesterol, and its oxidised derivatives. The determination in eight products should prove that the method works on several matrices. This article is the first one in a series of manuscript dealing with oxidised cholesterol derivatives. In subsequent articles we will focus on the content of COPs in fresh products and the effect of processing and storage on their levels in different types of food (dairy products, fishes, processed meat). We fully agree that the importance and market for plant-based products is growing. In our scientific work we concentrate at products of both plant and animal origin, in particular at lipid metabolites. The next stage of our research will be to study the oxidation of phytosterols in plant lipids.

Thank You for pointing out the interesting publications. We will certainly use them for further studies on phytosterol oxidation. Moreover, we you are very grateful for your comment highlighting the novelty of our research. Therefore, we have slightly changed the title of our article. Unfortunately, the problem of COPs is still poorly understood and there are no established maximum levels for these pollutants. I hope that series of publications prepared by our team will contribute to changing attitudes towards COPs and provide another argument for reasonable consumption of food products of animal origin.

Reviewer 3 Report

Comments and Suggestions for Authors

In this manuscript, squalene, cholesterol, and cholesteroloxidation products in the different food of animal origin analysed by using GC-TOF/MS. This is an interesting subjectand would be very useful if works.

It is straight forward work, the main question is enrichment step for cholesterol oxidation products and also its separation from impurities which also are high in case of cholesterol itself. The reason there is no such a method by now to analyse all these compounds together, is overlapping of cholesterol with its oxidation product and also the high amount of cholesterol compared with its oxidation products.

It could be very useful to have chromatogrms obtained for products in the manuscript. Did you check the purity of cholesterol peak? Please include its MS spectra as well. Otherwise the introduction part, methods and results and discussion parts are enough. However, the it should be clearly presented that how less amount of COPs are shown in the chromatograms as in the manuscript stated that for overcome of overlapping and having suitable cholesterol peak, the sample size should be less.

Author Response

Dear Reviewer,

Thank you for taking the time to review our article and for your valuable comments. Chromatograms and mass spectra have been added to the manuscript as supplementary material. The differences between cholesterol and the other compounds are quite large, so only the internal standard and cholesterol are visible on the main chromatogram (total ion chromatogram, TIC) whereas the individual COPs are revealed only by the monitoring of the respective ions. The purity of the cholesterol peak was checked at each of its positions. Additional ions, other than those characterising the cholesterol spectrum, were not visible. In  case of the eggs (which we wrote about in the article), two GC analyses of the same analytical sample were necessary: one to determine the COPs and one to determine the cholesterol content, which was mentioned in the text, as the peak of cholesterol was overloaded, and a quantitative analysis was impossible due to the too high error.

Reviewer 4 Report

Comments and Suggestions for Authors

This article dressed the quantitation of COPs in foods. The paper is well written. The method is solid. I suggest to publish this paper. There are only few place need to take attention.

1. In line 92-93 and Table 2. The LOD and LOQ was determined by standard compound in solvent as ng/micro L, which is the limit for the instrument. What are the LOD and LOQ for the method? For example,  can you quantitate squalene 0.25ng/uL in cheese?

2. In Table 1. the molecular weight Da is the average mass of the compound. However, the main diagnostic ions is the monoisotopic mass of the compound (which is detected by a mass spectrometer). The use of Da+/- 0.5 may confuse the reader, since the molecular weight is not the mass detected in a mass spectrometer (which is monoisotopic mass). You may dress in the manuscript that the window is 0.5 Da. Besides, since you use a TOF detector, please use accurate mass, for example for sq the diagnostic ion will be 410.39, and CH-TMS will be 386.35, etc.

3. In line 281, the hydrolysis temperature (room temperature) may be added.

4. the author may consider to explain, confirm, and discuss the COPs is form the sample, but not a artifact during sample preparation. 

5. The overal quality is good. Data is solid. A broad range of samples were tested. However, all the tested samples are fresh, have you ever test some presessed samples. The COPs may increase in precessed food. 

Comments on the Quality of English Language

Minor editing of English language required

Author Response

Dear Reviewer,

Thank you for taking the time to review our article and for your valuable comments. Below are the answers to your comments:

R: In line 92-93 and Table 2. The LOD and LOQ was determined by standard compound in solvent as ng/micro L, which is the limit for the instrument. What are the LOD and LOQ for the method? For example, can you quantitate squalene 0.25ng/uL in cheese?

A: Thank you for this suggestion. We have added additional columns to Table 2: LOD and LOQ for the method.

R: In Table 1. the molecular weight Da is the average mass of the compound. However, the main diagnostic ions is the monoisotopic mass of the compound (which is detected by a mass spectrometer). The use of Da+/- 0.5 may confuse the reader, since the molecular weight is not the mass detected in a mass spectrometer (which is monoisotopic mass). You may dress in the manuscript that the window is 0.5 Da. Besides, since you use a TOF detector, please use accurate mass, for example for sq the diagnostic ion will be 410.39, and CH-TMS will be 386.35, etc.

A: We agree that use of Da+/- 0.5 may be confusing for the Readers, so we have changed it. Instead we have expressed the masses of diagnostic ions with one decimal place.

R: In line 281, the hydrolysis temperature (room temperature) may be added.

A: We have added this information.

R: the author may consider to explain, confirm, and discuss the COPs is form the sample, but not a artifact during sample preparation. 

A: Thank you for raising this issue. We have tested different conditions of performing hydrolysis: different temperature conditions, without and with the presence of different antioxidants as well as different levels of antioxidant addition. The chosen procedure proved to be the most suitable. The lack of antioxidant addition caused an increase in the level of COPs in the samples. So did an increased temperature of hydrolysis. Hydrolysis under the selected conditions lasting from 8h to 26h did not result in changes in the level of COPs. The hydrolysis time adopted (20-22h) was due to the practical requirements of the work in the laboratory. Tests were also performed on the cholesterol standard itself and fortified samples. We assumed that the results of those individual preliminary experiments would not be interesting to the Readers and might only make the manuscript less clear. As the conditions adapted for the performed analytical procedure were optimal, we have added the following sentence to the manuscript: During the preliminary study, different hydrolysis times and temperatures as well as the addition of different antioxidants at different levels were investigated (data not shown). The presented procedure is the optimum obtained from our experiments.

R: The overal quality is good. Data is solid. A broad range of samples were tested. However, all the tested samples are fresh, have you ever test some presessed samples. The COPs may increase in precessed food. 

A: This manuscript deals with the inclusive method that we have tested on several matrices. It is the first of a series of publications. In the following steps we tested both fresh and processed products in three groups: dairy products / fish / fresh and processed meat (poultry and red meat), which we hope will be  published and available to the general audience soon.

Round 2

Reviewer 3 Report

Comments and Suggestions for Authors

As authors also are aware of drawback and weak point of this research work as stated in the reply to the comments "The differences between cholesterol and the other compounds are quite large, so only the internal standard and cholesterol are visible on the main chromatogram (total ion chromatogram, TIC) whereas the individual COPs are revealed only by the monitoring of the respective ions.", actually, it is not normal to evaluate a peaks, which are not visible in the chromatogram only just by ions and mass spectra. It could be better to emphasis this is issue in the abstract and discussion and also conclusion to give the readers clear picture of the developed method.

Author Response

Dear Reviewer,

Thank you sincerely for your comment. Following your suggestion, we have added a sentence about the limitations of our method in both the discussion and the summary. Due to the limited number of characters/words in the abstract, we have decided to leave it in its original form. However, we hope that we have sufficiently highlighted the problem.